# Engineered Graphene Quantum Dots as a Magnetic Resonance Signal Amplifier for Biomedical Imaging

**DOI:** 10.3390/molecules28052363

**Published:** 2023-03-03

**Authors:** Zhongtao Li, Guiqiang Qi, Guangyue Shi, Meng Zhang, Haifeng Hu, Liguo Hao

**Affiliations:** 1Department of Molecular Imaging, School of Medical Technology, Qiqihar Medical University, Qiqihar 161006, China; 2Animal Laboratory Center, Qiqihar Medical University, Qiqihar 161006, China; 3Department of MRI, The Second Affiliated Hospital of Qiqihar Medical University, Qiqihar 161006, China

**Keywords:** graphene quantum dots, magnetic resonance imaging, gadolinium, enhanced relaxivity

## Abstract

The application of magnetic resonance imaging (MRI) nano-contrast agents (nano-CAs) has increasingly attracted scholarly interest owing to their size, surface chemistry, and stability. Herein, a novel T1 nano-CA (Gd(DTPA)−GQDs) was successfully prepared through the functionalization of graphene quantum dots with poly(ethylene glycol) bis(amine) and their subsequent incorporation into Gd-DTPA. Remarkably, the resultant as-prepared nano-CA displayed an exceptionally high longitudinal proton relaxivity (r_1_) of 10.90 mM^−1^ s^−1^ (R^2^ = 0.998), which was significantly higher than that of commercial Gd-DTPA (4.18 mM^−1^ s^−1^, R^2^ = 0.996). The cytotoxicity studies indicated that the Gd(DTPA)−GQDs were not cytotoxic by themselves. The results of the hemolysis assay and the in vivo safety evaluation demonstrate the outstanding biocompatibility of Gd(DTPA)−GQDs. The in vivo MRI study provides evidence that Gd(DTPA)−GQDs exhibit exceptional performance as T1-CAs. This research constitutes a viable approach for the development of multiple potential nano-CAs with high-performance MR imaging capabilities.

## 1. Introduction

Magnetic resonance imaging is a well-established diagnostic technique renowned for its high temporal and spatial resolution and non-invasive monitoring capability [1,2]. Contrast agents (CAs) are employed to enhance the MRI effect, thereby augmenting diagnostic accuracy, specificity, and sensitivity [3,4,5]. Among the various CAs, Gd^3+^-diethylenetriamine-pentaacetic acid (Gd-DTPA) is an extensively adopted clinical agent due to its high thermodynamic stability [6,7]. However, Gd-DTPA has the deficits of low proton longitudinal relaxivity (r_1_), non-specificity, and a limited circulation time in vivo [8,9,10]. To address these limitations, Gd^3+^-complexed CAs, such as nanoparticles [11,12,13], liposomes [14,15], and dendrimers [16], have garnered increased attention due to their convenient surface modification potential, excellent stability, and long circulation time in vivo [17,18,19,20].

As a constituent of the nanomaterials community, graphene quantum dots (GQDs) have been deemed as a promising candidate in the realm of nano-CAs for MRI due to their salient attributes, such as low toxicity, low cost, and ease of functionalization [21]. In recent years, Gd^3+^-doped GQDs have elicited growing interest for their potential as dual modal magnetic resonance/fluorescence (FL) nano-CAs in biomedical imaging. For instance, Wang and colleagues developed multimodal imaging CAs by coating Gd_2_O_3_ nanoparticles with GQDs. The obtained Gd_2_O_3_/GQDs exhibited both high longitudinal relaxation (r_1_ = 15.995 mM^−1^s^−1^) and two-photon excitation property [22]. Another group, led by Lee, reported a straightforward hydrothermal approach for producing Gd^3+^-doped nitrogen-containing GQDs (NGQDs) for enhanced MR/FL dual imaging. The Gd-NGQDs displayed good biocompatibility and an increased r_1_ of 9.546 mM^−1^ s^−1^ [23]. Despite some progress being made in the preparation of Gd^3+^-doped GQDs, the preparation of Gd^3+^-based GQDs with an optimal balance between biosafety and high r_1_ values for biomedical imaging remains an ongoing challenge.

Polyethylene glycol (PEG) is a non-toxic polymer material that has been approved by the FDA [24]. This material has garnered significant attention in various biomedical applications due to its advantageous properties, such as biocompatibility, solubility, and modifiability [25]. Research has also indicated that PEG-functionalized materials can extend in vivo circulation time due to the resistance of PEGylated materials to protein biodegradation and adhesion [26].

To address these challenges, the authors developed a nanocomplex composed of GQDs and Gd^3+^ as a new type of nano-CA (Figure 1). In the developed nano-CAs, the GQDs were produced through a hydrothermal method and then functionalized with poly(ethylene glycol) bis(amine) to form a bridge with DTPA (diethylenetriaminepentaacetic acid dianhydride). The composite was then chelated with Gd^3+^ ions to form the nano-CAs, Gd(DTPA)−GQDs. The results of both in vitro and in vivo studies demonstrate that Gd(DTPA)−GQDs are novel, non-toxic, and high r_1_ value MRI CAs with the ability to enable FL imaging in vitro.

## 2. Results and Discussion

### 2.1. Synthesis and Structure of Gd(DTPA)−GQDs

In this investigation, the synthesis of water-soluble graphene quantum dots (GQDs) of uniform size was accomplished through the chemical oxidation and cutting of GO. The subsequent linking of Gd−DTPA to the GQDs through a PEG bridge was carried out as per Figure 1. The high-resolution transmission electron microscopy (TEM) images obtained illustrate that the average diameter of the GQDs was determined to be 3.64 ± 0.29 nm, exhibiting a highly crystalline structure with a lattice parameter of 0.21 nm (Figure 1a). Additionally, the atomic force microscopy (AFM) images depict that the thickness of the GQDs range between 0.86 and 2.54 nm (Figure 1b). After Gd(DTPA)−PEGylation, the size of the resulting Gd(DTPA)−GQDs was observed to have slightly increased to 4.15 nm, as evidenced in the TEM images (Appendix A), which suggests that the PEG has covered the GQDs [27]. The hydrodynamic (HD) sizes of both the GQDs and the Gd(DTPA)−GQDs were analyzed using dynamic light scattering (DLS) and were found to be 5.24 ± 0.38 nm and 9.1 ± 0.79 nm, respectively (Appendix A). Both the TEM and DLS measurements demonstrate a narrow size distribution. The zeta potential of the GQDs, as expected, was determined to be negative (−27.1mV) (Appendix A) due to the abundance of carboxyl groups on their surface [28]. After the DTPA−PEG−NH_2_ was grafted onto the surface of the GQDs to form Gd(DTPA)−GQDs, the zeta potential further increased to −23.8 mV, which can be attributed to the substantial presence of amine groups in PEG [29,30,31].

The Fourier transform infrared (FT−IR) spectra of both the GQDs and the Gd(DTPA)−GQDs are displayed in Figure 2a. The FT−IR spectrum of the initial GQDs exhibits the O−H stretching ~3440 cm^−1^, conjugated C=O ~1650 cm^−1^, and C−OH stretching ~1390 cm^−1^ [32]. After the surface modification process, significant decays in the peaks at 1650 and 3440 cm^−1^ were observed in the FT−IR spectrum of the Gd(DTPA)−GQDs, indicating the amidation of the carboxylic acid. Simultaneously, increases in the adsorption peaks at 1110 cm^−1^ were noted, which were attributed to the stretching of C−O−C in the PEG. A new peak at 2880 cm^−1^ was assigned to the N−H stretching vibration of the amine groups, providing evidence of the successful introduction of DTPA−PEG.

### 2.2. Optical Properties

The optical properties of the GQD samples are depicted in Figure 1c,d. The samples exhibit a pronounced absorption peak at 278 nm, corresponding to the transition of aromatic C−C bonds, and exhibit blue FL under 365 nm UV light irradiation (Figure 1c and insert) [33]. The photoluminescence emission spectra of the GQDs exhibit excitation-dependent features wherein the photoluminescence emission maximum shifts to higher energy and the emission intensity decreases as the excitation wavelength increases (Figure 1d). This shift may be a result of the optical selection of various surface defect states near the surface of the GQDs [34]. As depicted in Figure 2b, after further modification, the photoluminescence emission spectra of the GQD and Gd(DTPA)−GQD solutions were blue-shifted by 20 nm (under 350 nm laser excitation) at the maximum PL position, which could be attributed to Gd^3+^ complexation and the resulting energy transfer between Gd^3+^ and GQDs [35].

### 2.3. Assessment of the T1 Relaxivity

The concentration of Gd^3+^ in Gd(DTPA)−GQDs was quantified by the use of ICP−MS (2.75, atom. %). Subsequently, the suitability of Gd(DTPA)−GQDs as a nano−CA for MRI was evaluated by determining the relaxivity parameters [36]. The longitudinal relaxation rates (r_1_ values) of both Gd(DTPA)−GQD and Gd−DTPA samples were measured using the 0.5 T NMI20 Analyst NMR system at 37 °C. The results shown in Figure 3 indicate that the r_1_ rate of Gd(DTPA)−GQDs was 10.90 mM^−1^ s^−1^ (R^2^ = 0.998), which is significantly higher than that of commercial Gd−DTPA (4.18 mM^−1^ s^−1^, R^2^ = 0.996). The stronger relaxivity of Gd(DTPA)−GQDs compared with Gd−DTPA was tentatively attributed to the attachment of the Gd chelates to nanoparticles, prolonging the tumbling time and thus, increasing the relaxivity [37].

### 2.4. The Protein Adsorption and Stability of Gd(DTPA)−GQDs

Nanocarriers classified as “good” should exhibit low non-specific protein adsorption characteristic to prevent the immune system clearance and reduce the probability of phagocytic cell uptake, both of which are necessary to enhance the duration of circulation time [38,39,40]. To investigate the effect of PEG grafted onto graphene quantum dots (GQDs) on protein adsorption, an assay was performed using bovine serum albumin (BSA) as a model protein. The results, as displayed in Appendix A, indicate that GQDs had an adsorption of BSA of up to 12.1%, which is believed to have resulted from hydrogen bonding interactions and van der Waals interactions between GQDs and BSA [41,42,43]. However, after the GQDs were functionalized with PEG, the adsorption of BSA on the surface was significantly reduced to 2.9%. This outcome demonstrates that the modification of GQDs with PEG layers effectively inhibits BSA adsorption [44,45].

To assess the stability of Gd(DTPA)−graphene quantum dots (GQDs) in biological reagents, a sample consisting of 15 mg of Gd(DTPA)−GQDs was introduced into various solutions, including deionized (DI) water, saline, DMEM, and two variations of phosphate-buffered saline (PBS 1× and PBS 5×). As demonstrated in Appendix A, there was no noticeable alteration in the hydrodynamic size of the material throughout the observation period. The accompanying figure (Appendix A, insert) features photographs of Gd(DTPA)−GQDs in different biological solutions (DI water, saline, DMEM, PBS 1×, and PBS 5×), showing that the material remained stable without precipitation for 72 h. These results suggest that Gd(DTPA)−GQDs exhibit excellent dispersion in physiological solutions.

### 2.5. Biocompatibility and Cytotocivity

The cytotoxicity of graphene quantum dots (GQDs) and Gd(DTPA)−GQDs was evaluated on human normal pancreas (HPDE) cell lines and SW1990 cells using the CCK−8 assay. As depicted in Figure 4a and Appendix A, there was no noticeable cytotoxicity observed after 24 h of exposure to either of the nanoparticles at any of the tested concentrations. The hemolysis assay was also performed to ensure the feasibility of intravenous administration. Gd(DTPA)−GQDs of varying concentrations were incubated with red blood cells (RBCs), with saline and DI water serving as negative and positive controls, respectively. The hemolysis results, presented in Figure 4b and Appendix A, indicate that even at a high dose of 400 μg/mL, the hemolytic rate was below 5% and the RBCs remained intact, meeting the established standards for biomaterials hemolysis [46,47]. These findings demonstrate the excellent biocompatibility of Gd(DTPA)−GQDs.

### 2.6. In Vivo Safety Evaluation

The in vivo safety was evaluated by monitoring bodyweight, conducting histological examinations, and measuring blood chemistry indices. At seven days post−treatment, the bodyweight of the Gd(DTPA)−GQDs−treated group showed no significant differences compared to the saline group (Figure 5a). Furthermore, Figure 5b indicates that the levels of liver and kidney function (liver function markers: aspartate transaminase (AST), alanine aminotransferase (ALT), and total bilirubin (T−BIL); kidney function markers: blood urea nitrogen (BUN) and creatinine (Cr)) did not significantly differ from the saline group (*p* > 0.05) after treatment with Gd(DTPA)−GQDs, suggesting no observable harm to the primary functions of the liver and kidney. Hematoxylin and eosin (H and E) staining images did not reveal any noticeable pathological changes in the heart, liver, spleen, lung, kidney, and intestine after treatment with either Gd(DTPA)−GQDs or saline (Figure 5c). These results demonstrate the high biocompatibility of Gd(DTPA)−GQDs, making them a suitable candidate for further in vivo bioimaging applications.

### 2.7. Cellular Uptake

The cellular uptake efficiency of Gd(DTPA)−GQDs in SW1990 cells was evaluated utilizing fluorescence microscopy. As demonstrated in Figure 6, the Gd(DTPA)−GQDs fluorescence (represented by a blue color) accumulated within the cytoplasm of the SW1990 cells. With prolonged incubation, the blue signals within the cells increased, indicating that a greater number of Gd(DTPA)−GQDs were efficiently taken up by the cells over time (Appendix A). To further assess the intracellular MRI enhancement effect, the nano−CA−treated SW1990 cells were collected, suspended in centrifugation tubes, and then examined using a Philips 3.0T MRI scanner (Figure 5, insert). As depicted in Appendix A, the MR signal intensity increased with extended incubation time, demonstrating that more Gd(DTPA)−GQDs were successfully taken up by the cells over time.

### 2.8. Imaging in Tumor−Bearing Mice

The magnetic resonance (MR) properties of Gd(DTPA)−GQDs were subjected to testing. Intravenous injections of Gd−DTPA and Gd(DTPA)−GQDs were administered to SW1990 mice with tumors, each at a dose of 1 mg (1 mg of Gd^3+^ per kg). As depicted in Figure 7a, the location of the tumor (red circle) was distinctly visible 2 h post−injection. An examination of the MRI signal intensity images obtained from the Gd−DTPA and the Gd(DTPA)−GQD groups revealed a marked enhancement in the latter, likely due to its elevated r_1_ value (Figure 7b). These results indicate that Gd(DTPA)−GQDs hold potential as a T1−positive nano−contrast agent.

## 3. Materials and Methods

### 3.1. Materials

N-Hydroxysuccinimide (NHS, 98%), Diethylenetriaminepentaacetic acid dianhydride (DTPA, 98%), and N−(3−Dimethylaminopropyl)−N−ethylcarbodiimide hydrochloride (EDC·HCl, 99%) were offered by Macklin Biochemical Co., Ltd. (Shanghai, China). Poly(ethylene glycol) bis(amine) (PEG, M.W2000) was offered from Sigma-Aldrich (St. Louis, MO, USA). Gadolinium(III) chloride hexahydrate (GdCl_3_·6H_2_O, 99%) was provided by Aladdin Biochemical Technology Co., Ltd. (Shanghai, China). Graphene oxide (GO) was purchased from Nanjing XFNANO Materials Tech Co., Ltd. (Nanjing, China). Dulbecco’s Modified Eagle Medium (DMEM) and fetal bovine serum (FBS) were purchased from Gibco (Thermo Fisher Scientific, Waltham, MA, USA).

### 3.2. Synthesis of GQDs

The preparation of GQDs was based on a previously published method [48]. In brief, graphene oxide (100 mg) was combined with a mixture of 45 mL H_2_SO_4_ and 15 mL of fuming HNO_3_. The mixture was subjected to sonication for 30 min and then stirred at 120 °C for 1.5 h. After cooling, the solution was diluted with 200 mL of deionized water and neutralized with K_2_CO_3_ to achieve a pH range of 7–8. The resulting mixture was further purified by dialysis using a dialysis membrane (retained molecular weight: 3500 Da) for 24 h. The resulting GQDs were harvested through freeze−drying.

### 3.3. Synthesis of Gd(DTPA)−GQDs

The synthesis of Gd(DTPA)−GQD nanoparticles involved a series of steps. First, a solution composed of 30 mg of graphene quantum dots and 287 mg EDC·HCL in 50 mL DI water was stirred at a temperature of 37 °C for 3 h. Subsequently, 60 mg BocNH−PEG−NH_2_ and 172 mg NHS were introduced, and the pH was adjusted to 5.5 using 0.1 mM HCL before undergoing further agitation for 24 h at the same temperature. To eliminate the unresponsive precursors, the as-prepared solution was purified through 3500 Da molecular weight cutoff dialysis against DI water for 24 h. Then, 2 mL of trifluoroacetic acid was added to the resulting GQDs−PEG−NHBoc solution, stirred for 3 h at ambient temperature, and purified through dialysis against DI water to yield GQDs−PEG−NH_2_. GdCl_3_·6H_2_O (11.1 mg) was subsequently combined with an aqueous solution of DTPA (11.8 mg) for chelation and further modification to the GQDs−PEG−NH_2_ through reaction between the amino groups of PEG and the carboxyl groups of DTPA. The residual Gd^3+^ and DTPA were removed through 3500 Da cutoff dialysis for 12 h. Finally, the synthesized Gd(DTPA)−GQDs nanoparticles were subjected to freeze−drying to collect the samples.

### 3.4. Materials Characterization

The morphology of the GQDs was subjected to examination using transmission electron microscopy (TEM, FEI Talos F200S, New York, NY, USA) and atomic force microscopy (AFM, (Bruker Daltonics Inc. Multimode 8.0, Massachusetts, MA, USA). The Fourier transform infrared (FT−IR) spectra, UV−Vis absorption spectra, and FL spectra were obtained through utilization of FT−IR spectrometer (FT-IR 6800 JASCO, Marseille, France), Shimadzu UV−2450 spectrophotometer, and Hitachi 7000 fluorescence spectrophotometer, respectively. The hydrodynamic (HD) size and zeta potential of the samples were measured through the utilization of a nano ZS90 analyzer (Malvern Instruments Ltd., Worcestershire, UK), with the measurements being conducted at room temperature. The concentration of Gd^3+^ within the Gd(DTPA)−GQDs was confirmed through the application of inductively coupled plasma mass spectrometry (ICP-MS, Agilent 720 ES, Santa Clara, CA, USA).

### 3.5. Relaxivity Measurements

The T1 relaxivity values (r_1_) of Gd(DTPA)−GQDs in aqueous dispersions were determined through utilization of the NMI20 Analyst NMR System sourced from Niumag in China, with the measurements performed at 0.5 Tesla and 37 °C.

### 3.6. Cell Culture and Cellular Uptake Experiment

The human normal pancreas lines, HPDE, and human pancreatic cancer cell lines, SW1990, were procured from the Cell Bank of the Shanghai Institute of Cell Biology (China). They were sustained in a culture medium composed of DMEM containing 10% FBS and antibiotics (1% penicillin/streptomycin).

To measure cellular uptake, SW1990 cells were cultured in 12−well plates and allowed to grow overnight. Subsequently, Gd(DTPA)−GQDs (100 µg/mL) were introduced into designated wells for varying durations (0, 12, and 24 h). The cells were then thoroughly rinsed with phosphate-buffered saline (PBS) and fixed utilizing a 4% formaldehyde solution for 20 min. The resultant cells were observed using confocal laser scanning microscopy (CLSM).

To validate the enhancement effect of Gd(DTPA)−GQDs as MRI nano−CAs within cells, SW1990 cells were cultured in 6−well plates. Gd(DTPA)−GQDs were incorporated into the culture medium at varying intervals (0, 12, and 24 h). The cells were then harvested by centrifugation and fixed with a 1.0 % wt agarose solution. The MRI experiments were conducted utilizing a clinical 3.0 T MRI scanner (Philips, The Netherlands). The parameters for T1−weighted (T1W) imaging were set as follows: matrix = 256 × 256, layer thickness = 5 mm, FOV (field of view) = 50 × 50 mm, TR = 650 ms, TE = 10 ms, echo time (TE) = 10 ms. Raw data were transmitted to a remote computer for analysis.

### 3.7. BSA Adsorption and Colloidal Stability

The amount of bovine serum albumin (BSA) adsorption was prepared in accordance with a published study [49]. BSA was completely dissolved in PBS (pH 7.4, 0.3 mg/mL). Subsequently, 15 mg of both GQDs and Gd(DTPA)−GQDs were dispersed into a 10 mL solution of BSA and the mixture was stirred at 37 °C for 5 h at a shaking rate of 120 rpm. The supernatant was collected through centrifugation. Finally, the final concentration of BSA was determined through a Coomassie brilliant blue stain assay.

To assess the colloidal stability, Gd(DTPA)−GQDs were dispersed in deionized water, saline, DMEM medium, PBS 1×, and PBS 5× at a certain concentration (2 mg/mL), respectively. The samples were kept in an incubator at 37 °C and the alteration in hydrodynamic (HD) size were monitored through dynamic light scattering (DLS) measurements for a period of up to 72 h.

### 3.8. Biocompatibility

The cytotoxicity impact of Gd(DTPA)−GQDs was appraised against HPDE cells and SW1990 cells by CCK−8 assay. To initiate the process, the cells were cultured in separate 96−well plates at a concentration of 8 × 10^3^ cells/well and allowed to grow in DMEM medium overnight. Subsequently, varying concentrations of Gd(DTPA)−GQDs (0, 5, 50, 100, 150, and 200 μg/mL) were added to the wells for a 24 h duration. The viability of the cells was finally assessed through CCK−8 assay.

To delve further into the actual condition, the compatibility of Gd(DTPA)−GQDs with human red blood cells was assessed using samples procured from volunteer donors [50,51]. The red blood cells were initially collected through centrifugation and subjected to purification via three successive washes with saline. Subsequently, a mixture of 2 mL of the RBC suspension (4% *v*/*v*) was combined with: (a) 2 mL of deionized water as a reference positive control; (b) 2 mL of saline as a reference negative control; and (c) 2 mL of Gd(DTPA)−GQD suspensions in saline at various concentrations (5, 50, 100, 200, and 400 µg/mL). The resulting mixtures were then incubated at 37 °C for a duration of 3 h prior to being subjected to centrifugation. The absorbance of the supernatant layers at 541 nm was measured using a UV−Vis spectrophotometer. The hemolysis percentage was calculated using the following equation (repeated three times for each respective sample).
(1)$$\mathrm{Hemolysis}(\%)=\frac{\left(A_{Sample}-A_{Negative\;Control}\right)}{\left(A_{Positive\;Control}-A_{Negative\;Control}\right)}\times 100\%$$

The biotoxicity of Gd(DTPA)−GQDs was assessed using Balb/c-nu mice (male), with approval from the Animal Ethics Committee of Qiqihar Medical University (No. QMU−AECC−2021−100). Twelve mice, weighing between (18–22g), were randomly divided into two groups (*n* = 6). The mice were administered intravenous injections of either 100 μL of normal saline or Gd(DTPA)−GQDs (1 mg/mL). After a seven day period, approximately 0.5 mL of blood was collected from each mouse for biochemical analysis prior to euthanasia. The main organs were then harvested and subjected to histopathological examination (Leica−DM4B digital microscope, Hesse, Germany).

### 3.9. In Vivo Proof-of-Concept Study

An investigation into the MRI enhancement effect of the synthesized nano−CAs was carried out in a nude mouse tumor model. Specifically, SW1990 cell (1 × 10^7^) suspensions were subcutaneously implanted into the right armpits of Balb/c-nu mice. The SW1990 tumor-bearing mice were then subjected to intravenous injection with Gd(DTPA)−GQDs for T1−weighted MRI (1 mg of Gd^3+^ per kg). The solutions were sterilized by filtration through membranes (pore size 0.22 μm). The MRI study was conducted using a 3.0 Tesla clinical MRI scanner (manufactured by Philips) equipped with an 8-channel carotid wall imaging special phased array coil. The Balb/c−nu mice were anesthetized using chloral hydrate through intraperitoneal injection. T1−weighted MRI scans were obtained in cross−sectional and coronal views with the following parameters: TR/TE = 650/10 milliseconds, layer thickness = 3 mm, layer spacing = 2 mm, FOV = 100 × 100 mm, matrix = 192 × 192. MR images were analyzed before injection and 2h after injection. The signal intensity (SI) of the tumor tissue was measured in approximately the same slice with a consistent diameter for each region of interest (ROI).

### 3.10. Statistical Analysis

The statistical data were investigated using SPSS 20.0 software with a Student’s *t*-test. All findings were expressed as the mean ± standard deviation. Here, a *p*-value of < 0.05 was considered indicative of statistical significance.

## 4. Conclusions

In this work, a Gd(DTPA)−GQDs−based T1−positive nano−CA (Gd(DTPA)−GQDs) was developed through a surface modification process that involved integrating Gd−DTPA and GQDs via a PEG bridge. The results showed that PEG−coated GQD nanoparticles had low nonspecific protein adsorption, high colloidal stability, and exceptional biocompatibility. Furthermore, in terms of MRI contrast agent capabilities, Gd(DTPA)−GQDs exhibited a high longitudinal relaxivity rate of 10.90 mM^−1^ s^−1^, which was significantly higher than that of the commercial Gd−DTPA (4.18 mM^−1^ s^−1^). In vivo assays confirmed that Gd(DTPA)−GQDs were suitable as a T1 contrast agent. It is anticipated that this type of nano−contrast agent possesses the potential to extend the applications of nanotechnology in non-invasive diagnosis and therapy.

## Data Availability

Data will be made available on request.

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
