# Peer review of "Engineered Graphene Quantum Dots as a Magnetic Resonance Signal Amplifier for Biomedical Imaging"

_molecules, 2023, doi:10.3390/molecules28052363_

Round 1

Reviewer 1 Report

The manuscript is focused on the functionalization of graphene quantum dots (GQDs) and their subsequent incorporation into Gd3+-diethylenetriamine-pentaacetic acid (Gd-DTPA). As prepared Gd(DTPA)-GQDs nanocomplex were evaluated as potential nano-contrast agent (nano-CA). It was proved that nanocomplex exhibited extremely high longitudinal relaxivity rate in comparison with the commercial Gd-DTPA. Moreover, in vivo assays confirmed that Gd(DTPA)-GQDs nanocomplex was suitable as a T1 contrast agent. The used methodology is described in sufficient details. The characteristics of the prepared GQDs, Gd-DTPA and their nanocomplexes, as well results are original, valuable and well described. There is some minor inaccuracies (corrections to minor errors and text editing) that need fixing.

Line 17, 51, etc. throughout the text – “in vivo” should be in italic

Line 338 - Furthermore, In terms – “in” should be small caps

Reviewer 2 Report

The manuscript entitled "Engineered Graphene Quantum Dots as a Magnetic Resonance Signal Amplifier for Biomedical Imaging" reported a potential application of a prepared MIR contrast material. This new material was demonstrated to be biocompatible and of high longitudinal proton relaxivity (r1).  This material is well characterized and has good properties for amplifying MRI signals. Before further consideration for publication, the following concerns should be addressed.
1.    Please give more details on the mechanism of this material in amplifying the MRI performance, rather than just showing the result. The mechanism will help readers of interest design more effective materials for practical use.
2.    Please explain the result of the biocompatibility test in the abstract.
3. In the abstract, please include a quantified result comparison of the material's performance in amplifying the MRI signal.
4.    Is this material of high specificity in MRI performance? If yes, please explain the target tissue or organ, and further give the reason for this performance.
5.    In line 44, it should be "an increased R1 of " rather than "a increased r1 of".  Please check through the full text for similar grammar problems.
6.    In line 52, "in vivo circulation times" or "in vivo circulation time"? Please check.
7.    Since the material is of high FL property, can it be imaged with good FL inside  target cells?
8.    Please add annotation about the concentration of this material used for the preparation of Figure 1c and Figure 1d.
9.    Please check whether you are referring to the right picture in line 101.
10.    There are no units for the y-axis of Figure 1c and Figure 1d.
11.    Please specify the concentration of the measured materials in preparation of Figure 2b.
12.    In line 121, please explain the mechanism for extending the T1 relaxivity.
13.    Add error bars to Figure 3.
14.    Check your writing in line 133.
15.    Explain why the PEG layers can inhibit BSA adsorption.
16.    In line 160, please give more details on the blood chemistry indices. What does it refer to?
17.    In line 196, explain why you use the concentration of 1 mg of Gd3+ per kg.
18.    In line 197, please specify the specificity of this material, and further give the reason.
19.    Check your writing in line 295.

Reviewer 3 Report

The authors have written an article entitled “Engineered Graphene Quantum Dots as a Magnetic Resonance Signal Amplifier for Biomedical Imaging”. The work reported in this manuscript is interesting and very well-presented. The article has some grammatical and sentence errors, and the language organization needs to be improved. The authors have described the concept to a reasonable extent but the manuscript still needs some Minor corrections before publishing in the Molecules.

I advise the authors to consider the following points when revising their manuscript.

1.      Minor punctuation revision is required in the manuscript.

2.      The manuscript needs to be checked for typographical/ grammatical, superscript, and subscript errors.

3.      The abstract is to be rewritten especially the first 2 lines there are words like garnered doesn’t look like scientific terminology. Revise it

4.      In the abstract authors did not mention in vitro analysis and results, can explain?

5.      Keywords do not match the manuscript example in vivo to be incorporated

6.      In statistical analysis, authors concluded that all the results were expressed as the mean ± standard deviation, in Figure 4 thy did not show? why?

7.      As can be seen from Figure 4 (a), results are to correlate with raw data as a figure showing no difference at any concentrations which is a bit strange to observe, authors can send us raw data for cell cytotoxicity assay.

8.      Mainly figure 4 is to be separated and results should be explained separately in vitro and in vivo

9.      The authors did not mention ethical permission and clearance.

10.  What statistical software was used in analyzing the data did not mention?

Round 2

Reviewer 2 Report

I believe it can be published just after moderate Enlgish polishing.